

# Exosomes from uterine fluid promote capacitation of human sperm

Renbin Deng[1,*], Zhao Wu[2,*], Chaoyong He[1], Chuncheng Lu[1], Danpeng He[1], Xi Li[1], Zhenling Duan[3] and Hui Zhao[1]

[1] Department of Urology, The First Affiliated Hospital of Kunming Medical University, Kunming, Yunnan, China
[2] Department of Genetics, The First Affiliated Hospital of Kunming Medical University, Kunming, Yunnan, China
[3] Department of Gynecology, The First Affiliated Hospital of Kunming Medical University, Kunming, Yunnan, China

[*] These authors contributed equally to this work.

Corresponding authors
Zhenling Duan,
zhenlingduan@126.com
Hui Zhao, ydyyzh@126.com

## ABSTRACT

**Background.** Extracellular vesicles (EVs) are membrane-bound vesicles containing various proteins, lipids, and nucleic acids. EVs are found in many body fluids, such as blood and urine. The release of EVs can facilitate intercellular communication through fusion with the plasma membrane or endocytosis into the recipient cell or through internalization of the contents. Recent studies have reported that EVs isolated from human endometrial epithelial cells (EECs) promote sperm fertilization ability. EVs from uterine flushing fluid more closely resemble the physiological condition of the uterus. However, it is unclear whether EVs derived directly from uterine flushing fluid have the same effect on sperm. This study aimed to research the effect of EVs from uterine flushing fluid on sperm.

**Methods.** EVs were isolated from the uterine flushing fluid. The presence of EVs was confirmed by nanoparticle tracking analysis (NTA), Western blot, and transmission electron microscopy (TEM). EVs were incubated with human sperm for 2 h and 4 h. The effects of EVs on sperm were evaluated by analyzing acrosome reaction, sperm motility, and reactive oxygen species (ROS).

**Results.** The EVs fractions isolated from the uterine fluid were observed in cup-shaped vesicles of different sizes by TEM. All isolated vesicles contained similar numbers of vesicles in the expected size range (30–200 nm) by NTA. CD9 and CD63 were detected in EVs by western blot. Comparing the motility of the two groups incubated sperm motility significantly differed at 4 h. The acrosome reactions were promoted by incubating with EVs significantly. ROS were increased in sperm incubated with EVs.

**Conclusion.** Our results showed EVs present in the uterine fluid. Acrosome reactions and ROS levels increased in human sperm incubated with EVs. EVs from uterine fluid can promote the capacitation of human sperm. The increased capacitation after sperm interaction with EVs suggests a possible physiological effect during the transit of the uterus.

## INTRODUCTION

After ejaculation, sperm must travel through the female genital tract to combine with the oocyte to form a zygote. After entering the female genital tract, the sperm count in semen ranges from $10^8$–$10^9$ initially and is then reduced to $10^2$–$10^3$ before reaching the oviduct. Despite the fact that the interaction time between sperm and female genital secretions is limited, the time is essential for sperm capacitation (*De Jonge, 2017*). Early studies suggest that the uterus and oviduct might synergistically accelerate sperm capacitation (*Bedford, 1970*; *Hunter & Hall, 1974*); thus, the female reproductive tract may optimize sperm by regulating sperm capacitation.

Extracellular vesicles (EVs) are phospholipid bilayer-enclosed entities that are produced by eukaryotic and prokaryotic cells. EVs carry and deliver various regulatory molecules, including proteins, microRNAs, and lipids, and play key roles in cell–cell communication (*Machtinger, Laurent & Baccarelli, 2016*). *Griffiths et al. (2008)* first discovered uterine exosomes in the uterine fluid of estrous mice, and uterine EVs are potentially responsible for sperm capacitation, membrane stabilization, and eventual maturation *via* miRNA transfer in mice (*Godakumara et al., 2022*). Recent studies have shown that uterosome-like vesicles derived from cultured human endometrial epithelial cells (EECs) promote sperm capacitation (*Franchi et al., 2016*; *Murdica et al., 2020*). However, these findings were generated by EVs derived from endometrial adenocarcinoma cells and do not fully represent the physiological properties of EVs in uterine fluid. EVs isolated from uterine fluid are therefore more representative of the internal environment of the female reproductive tract and thus more suitable for functional analysis.

In this study, we applied the method recently developed by *Luddi et al. (2019)* to isolate EVs from uterine fluid, and evaluated their effects on sperm. We also demonstrated that EVs derived from uterine fluid act on sperm directly and promote sperm capacitation.

## MATERIALS AND METHODS

### Ethical approval

The study was approved by the Ethical Committee of the First Affiliated Hospital of Kunming Medical University, China, and all of the participants provided written informed consent. This study was designed and conducted according to the Declaration of Helsinki.

### Sperm preparation

Semen was collected by masturbation from healthy donors ($N = 3$, age 25–35 years) who had been abstinent for 2–7 days. The semen parameters of donors were normal according to the World Health Organization (WHO) guidelines (*Björndahl & Kirkman Brown, 2022*). Semen was liquefied for 30 min at room temperature, and spermatozoa were isolated from the semen by discontinuous Percoll gradient centrifugation. Sperm were resuspended in Biggers-Whitten-Whittingham (BWW) medium (Solarbio, Beijing, China) and maintained at 37 °C in an incubator with 5% $CO_2$ in compressed air and high humidity. In the experiments on sperm capacitation, we used a non-capacitating BWW medium (NC-BWW, Hepenbio, China), which did not contain human serum albumin or $HCO_3^-$.

## Collection and processing of uterine flushes

Ten female volunteers with proven fertility were recruited in this study. Inclusion criteria were: (1) 25–45 years of age, (2) proven fertility, (3) no HPV infection, and (4) no cervical or uterine inflammation. The uterine flush fluid was collected at the Gynecology Department of the First Affiliated Hospital of Kunming Medical University. Briefly, the uterus was flushed with a disposable catheter filled with 40–50 ml of normal saline before hysteroscopic surgery. Samples were immediately collected from the catheter, and the supernatant was filtered through a 0.22-µm filter to remove cells and cellular debris. The supernatant was then transferred to an Amicon Ultra-15 10 K centrifuge filter (Merck Millipore, Rahway, NJ, USA), centrifuged at 5,000 g for 10 min at 4 °C to concentrate it to 2 ml, and stored at −20 °C.

## EV isolation

EVs were isolated using ExoQuick TC™ (SBI, CA, USA) according to the manufacturer's instructions (*Alvarez, 2014*). Briefly, the supernatant was thoroughly mixed with ExoQuick TC™ at a volume/volume ratio of 2:1, and the mixture was maintained at 4 °C for 12–16 h, followed by centrifugation of 16,000 g at 4 °C for one hour. The supernatant was then discarded. The resulting EV pellet was resuspended in 100 µl of sterile PBS and preserved at −80 °C. EVs from 10 donors were pooled and then incubated with sperm from three different donors.

## Transmission electron microscopic analysis of EVs

The quality of isolated EVs was assessed by transmission electron microscopy (TEM) as described previously (*Yang et al., 2019*).

## Nanoparticle tracking analysis

Nanoparticle tracking analysis (NTA) was performed using a ZeTaView Particle Metrix instrument (*Szatanek et al., 2017*). Briefly, the samples were first diluted with PBS to achieve 20–120 nanoparticles per frame and/or $10^7$–$10^9$ nanoparticles/ml. Three 30-second videos were then recorded at 30 frames per second for each sample. The concentration, mean size and SEM of the nanoparticles in each sample were calculated.

## Western blot analysis

EVs were extracted and subjected to protein quantification using a BCA kit and stored at −80 °C. Samples selected for electrophoresis were prepared in SDS-PAGE sample loading buffer and heated for 5 min at 100 °C. Ten micrograms of protein were loaded per lane on 10% polyacrylamide gels and transferred onto polyvinylidene fluoride (PVDF) membranes. Blots were then blocked for 2 h at room temperature and incubated with anti-CD63 (1:500, 30–55 kDa; 25682-1-AP, Proteintech, Chigago, IL, USA) and anti-CD9 (1:20,000, 23–30 kDa; 20597-1-AP, Proteintech, Chicago, IL, USA) primary antibodies overnight at 4 °C. Non-specific binding of antibody was removed by washing three times with TBST for 10 min each before incubation with goat anti-rabbit IgG H&L (HRP, 1:10,000) for one hour at 4 °C. The membrane was again washed three times with TBST for 10 min chemiluminescence was detected using an ECL kit (*Campoy et al., 2016*; *Kowal et al., 2017*).
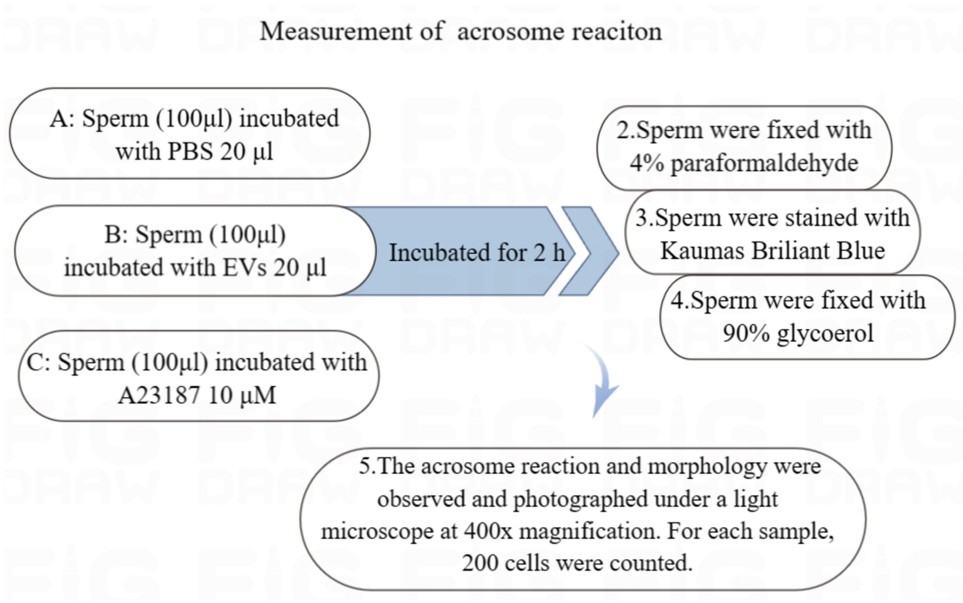

**Figure 1** **Flowchart for the measurement of acrosome reaction.** (A) Sperm incubated with PBS (unincubated with EVs). (B) Sperm incubated with EVs. (C) Sperm incubated with A23187.

## Sperm motility

Sperm were incubated with EVs at 37 °C for 2 h and 4 h. Curvilinear velocity (VCL) and percentage of motile sperm were determined using a computer-assisted sperm analysis system (Sperm Class Analyzer; Microptic, Barcelona, Spain).

## Measurement of the acrosome reaction

The sperm were equally divided into three groups and incubated with 20 μl of PBS, 20 μl of EVs, or 10 μM calcium ionophore A23187 (Sigma-Aldrich, St. Louis, MO, USA) for 2 h at 37 °C, respectively. The sperm from all of the groups were fixed in 4% paraformaldehyde for 25 min at room temperature and then washed twice with PBS (*Xu et al., 2021*).

The fixed sperm were stained with 0.22% Kaumas Brilliant Blue G250 (Sigma Aldrich, St. Louis, MO, USA) for 5 min to visualize sperm heads. After rinsing with distilled water, the sperm were allowed to dry and were fixed with 90% glycerol. The acrosome reaction and morphology were observed and photographed under a light microscope at 400x magnification. We counted 200 cells from each sample (Fig. 1) (*Murdica et al., 2019*).

## Measurement of ROS

Sperm were incubated with EVs for 2 h at 37 °C, centrifuged at 1,400 r/min for 8 min, and the supernatant aspirated. After that, 2′, 7 Dichlorofluorescein diacetate (DCFH-DA, MedChemExpress, Monmouth Junction, NJ, USA) was added and incubated with sperm for 20 min at 37 °C. Sperm were centrifuged twice to remove DCFH-DA, and resuspended in 100 μl of PBS solution and incubated at room temperature in the dark for 20 to 30 min. Fluorescence intensity was determined using flow cytometry (ACEA Biosciences, San Diego, CA, USA).

## Statistical analysis

Sperm from three donors were used to conduct experiments, and experiments with each donor's sperm were replicated three times. We adopted one-way analysis of variance (ANOVA) followed by Tukey's *post hoc* test to determine the differences among groups using SPSS 22.0 software. $p < 0.05$ was considered to be statistically significant.

# RESULTS

## Characteristics of EVs in uterine flushes

TEM analysis confirmed the presence of EVs in all 10 of the uterine douche samples collected in this study. As shown in representative electron microscopic images (Figs. 2A, 2B), the fraction isolated from the uterine douche showed cup-shaped vesicles of different sizes. All of the vesicles showed a honeycomb-like double-membrane structure with diameters ranging from 30 nm to 200 nm, a typical feature of EVs (*Franchi et al., 2016*).

Next, further characterization of EVs using NTA showed that all of the isolated vesicles contained similar numbers of vesicles in the expected size range (30–200 nm) (Figs. 2C, 2D) (*Szatanek et al., 2017*), and that EVs from uterine fluid showed characteristics similar to EVs from EECs cultured in vitro. We detected CD9 and CD63 in EVs from different donors by western blot analysis (Fig. 2E) (*Campoy et al., 2016*).

## Effect of EVs on the motility of human sperm

Sperm incubated with EVs for 2 h were measured using an SQA-V automated semen analyzer, and the data were analyzed using a computer-assisted semen analysis system. We herein considered the percentage of sperm with forward and non-forward motion as an indicator of sperm motility. Our results revealed that sperm motility did not differ at 2 h, but was significantly different at 4 h (Fig. 3).

## Effect of EVs on sperm capacitation

It has been demonstrated that uterine fluid promotes sperm capacitation and the acrosome reaction, and we herein assessed sperm capacitation by the level of acrosome reaction. After staining with Kaumas Brilliant Blue and observation by light microscopy, sperm heads with an intact acrosome stained blue, and the regions of the sperm heads after the reaction remained unstained (*Lu et al., 2002*). We observed acrosome reactions in every group (Fig. 4).

The percentage of unstained sperm acrosomes in group A was significantly lower than in groups B and C, showing that the percentage of acrosome-reacted sperm increased significantly after incubation with EVs or A23187 (Fig. 5). These data suggest that uterine fluid EVs modulate the physiological state of sperm and increase their energetic capacity.

## EVs increase levels of ROS in human sperm

The ROS content of the sperm in each group was quantified by measuring the fluorescence intensity using flow cytometry. DCFH-DA is a cell-permeable dye that is hydrolyzed intracellularly to DCFH, and oxidation of DCFH by the action of intracellular ROS transforms the molecule into its highly fluorescent form, DCF (*Rajneesh Pathak et al.,*

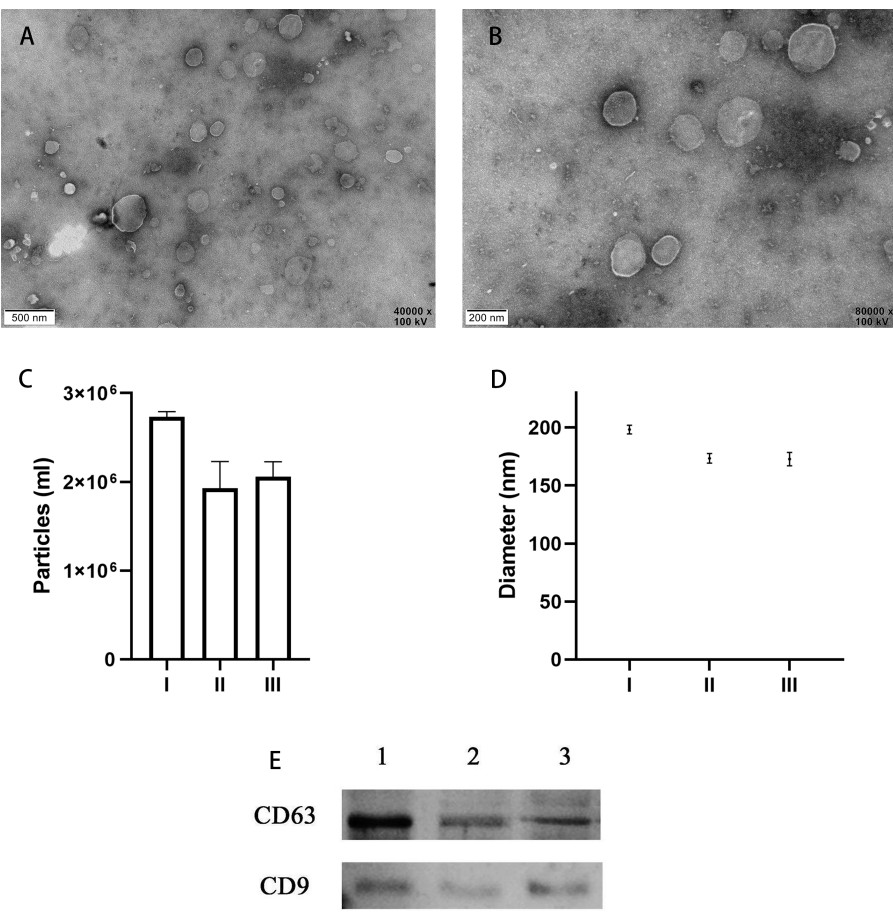

**Figure 2 Characteristics of EVs isolated from uterine fluid.** (A–B) Representative TEM images of EVs isolated from uterine fluid at 40,000 × (A) and 80,000 × (B) magnification (scale bars are 0.5 μm in A and 0.2 μm in B). (C–D) Quantification of EV particle number (C) and size (D) by NTA. (C) The data are presented as the mean and SEM of three experimental data points. The EVs particles are expressed as the mean ± SEM of three repeated measurements (I, 2.73 × 106 ± 0.03; II, 1.93 × 106 ± 0.17; III, 2.06 × 106 ± 0.07 mL). (D) The diameter of EVs in the three groups (I, 198.3 ± 2.16; II, 173.4 ± 2.40; III, 172.7 ± 2.59 nm; NS, $p > 0.05$). (E) CD63 and CD9 were used to label EVs extracted from uterine flushes. 1, 2, and 3 represent uterine flushes of EVs from different donors. The results showed that CD63 and CD9 markers could be detected in EVs extracted from uterine flushes.

*2017*). When we applied DCFH-DA to detect cytosolic ROS production in sperm, we observed that the fluorescence intensity increased significantly after sperm were incubated with EVs from uterine flushes, indicating an increase in ROS. This result suggests that EVs increased the ROS levels of sperm (Fig. 6).

## DISCUSSION

In animals exhibiting *in vivo* fertilization, sperm migrate toward the female's primary reproductive organ after mating and fuse with the oocyte, with tens or hundreds of millions of sperm found in the ejaculate. Therefore, most species possess a rigorous screening process to ensure that high-quality sperm eventually combine with the egg.

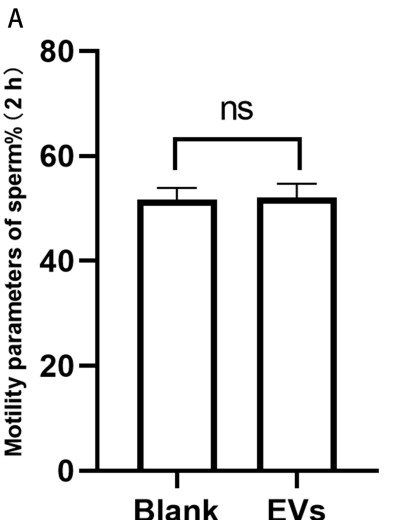
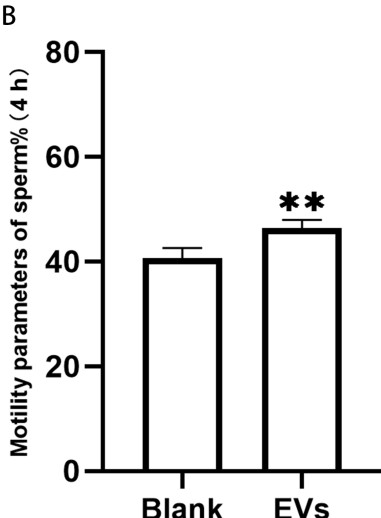

**Figure 3** **Motility of sperm after incubation with EVs.** (A) represents sperm motility after two hours of incubation with EVs. (B) represents sperm motility after four hours of incubation with EVs. Blank, sperm unincubated with EVs; EV group, sperm incubated with EVs. Three biological replicates were performed. The data represent the mean and SEM of the three experimental data points. (A) 51.63% ± 1.32% *vs.* 52.10% ±1.51%, $p > 0.05$; (B) 40.6% ± 1.13% *vs.* 6.43% ± 0.86%, **$p < 0.01$.

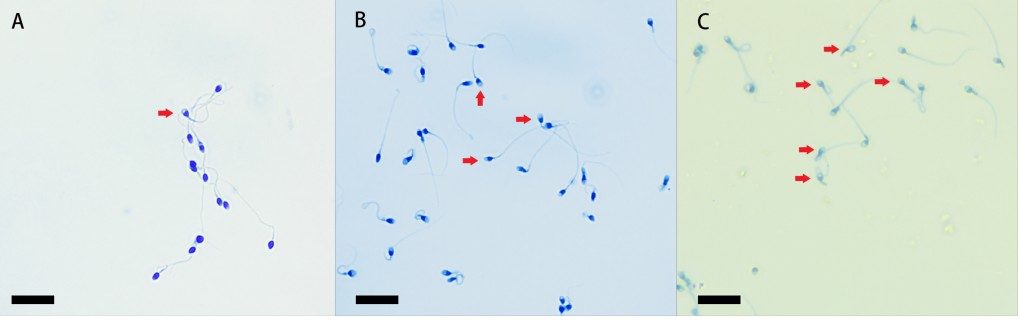

**Figure 4** **Coomassie brilliant blue staining of sperm acrosomes.** The heads of sperm with intact acrosomes are shown as blue, and the area of the sperm head after the reaction was unstained. The acrosome-reacted sperm are indicated by red arrows, and three groups were stained with Coomassie Brilliant Blue. Original magnification, 400 ×; scale bar, 20 μm. (A) sperm unincubated with EVs; (B) sperm incubated with EVs; (C) sperm incubated with A23187.

Sperm enter the female reproductive tract and undergo a long and tortuous journey through the vagina, cervix, uterus, uterine tubal junction, and oviducts before only one oocyte is usually successfully fertilized by one spermatozoon. This implies that the sperm will interact with and be affected by the uterine environment during transport through the uterus. We demonstrated that EVs isolated from uterine fluid rapidly interact with sperm and thereby enhance human sperm fertilizing ability (*Eisenbach & Giojalas, 2006*).

Earlier studies suggested that the uterus and oviduct synergistically accelerate sperm capacitation (*Hunter, 1969*; *Hunter & Hall, 1974*). For example, it was found in wild boars

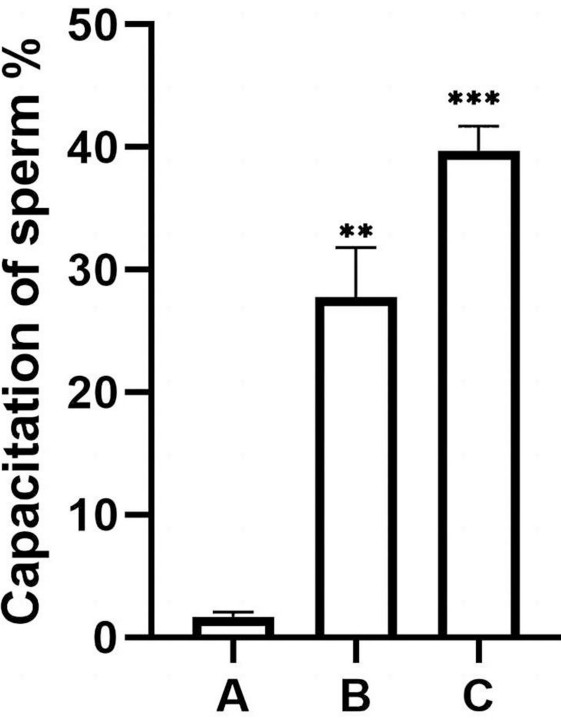

**Figure 5** **Percentage of acrosomally reacted sperm in each group.** (A), sperm unincubated with EVs; (B), sperm incubated with EVs; (C), sperm incubated with A23187. The results showed that the acrosome reaction rate increased significantly in group B and group C compared with group A. Three biological replicates were performed. The results are expressed as the mean $\pm$ SEM of three experiments (A, 1.67 $\pm$ 0.22; B, 27.75 $\pm$ 2.34; C, 39.73 $\pm$ 1.16) ($^{**}p < 0.01$ and $^{***}p < 0.001$).

that sperm could be temporarily stored in the uterus before entering the oviduct (*Hunter & Rodriguez-Martinez, 2004*), and Fusi and colleagues demonstrated that incubation of EECs with sperm enhances their fertilizing capacity. The number of sperm penetrating hamster oocytes was also significantly increased after co-culture of EECs and sperm (*Fusi et al., 1994*).

*Franchi et al. (2016)* demonstrated that even 15 min of sperm incubation with EVs was sufficient to increase the level of active cells up to 1.5-fold. This suggests that EVs rapidly interact with the sperm plasma membrane to modulate the physiological state of sperm and improve their energetic capacity, even after a brief period of rapid uterine transit.

As sperm motility did not differ with EVs at 2 h, we speculated that this incubation period was insufficient and therefore increased the incubation time. After 4 h of incubation, our results then revealed that sperm motility was significantly enhanced. We thus demonstrated that EVs from uterine fluid promoted sperm capacitation, and after capacitation sperm motility was subsequently slightly enhanced. Data from *Sáez-Espinosa et al. (2020)* showed that one hour of capacitation sufficed to recover a sperm subpopulation with high levels of motility and viability, and revealed that sperm recovery after swim-up was approximately 15% after 1 h and 4 h. In conclusion, our findings indicated that EVs promoted sperm

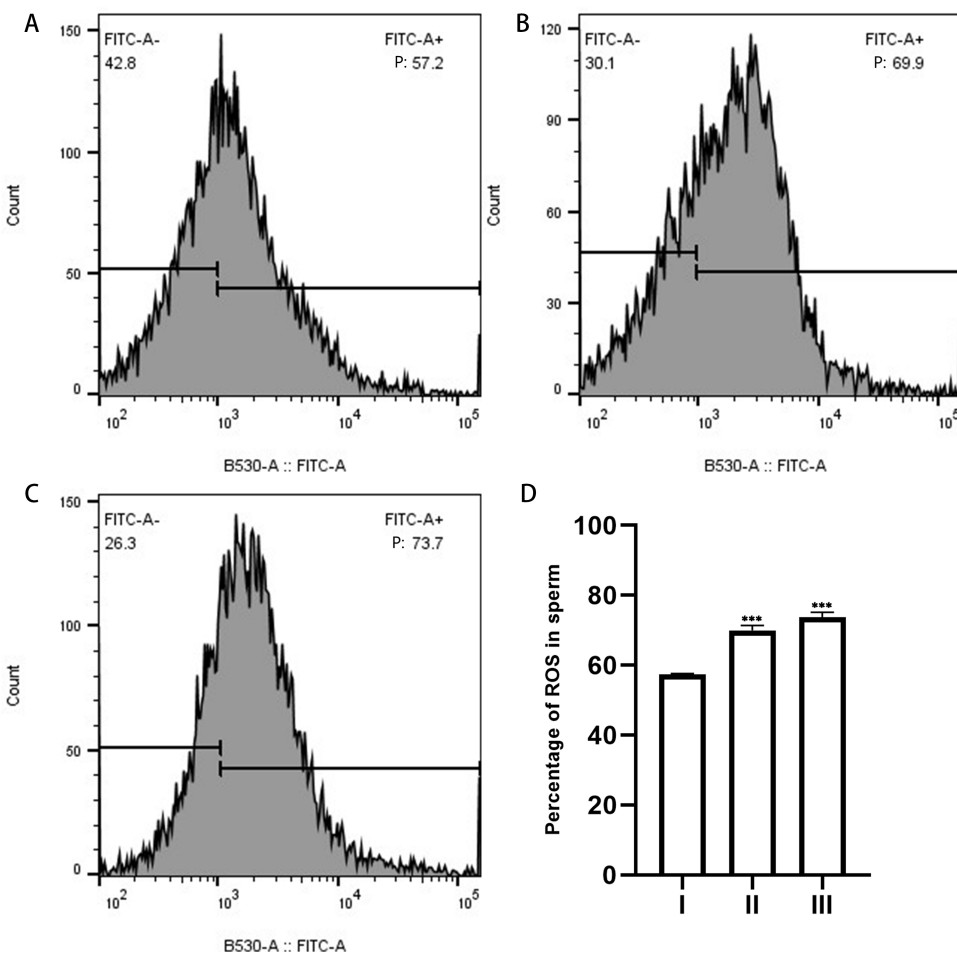

**Figure 6** **ROS in sperm as measured by flow cytometry.** (A), sperm unincubated with EVs; (B), sperm incubated with EVs; (C), sperm incubated with A23187. "P" in the figure indicates the percentage of DCFH-DA fluorescence; increased fluorescence intensity denotes increased production of ROS. (D) The results showed that the levels of ROS increased significantly in groups B and C compared with group A. Three biological replicates were performed. The results are expressed as the mean $\pm$ SEM of three experiments. I, sperm unincubated with EVs; II, sperm incubated with EVs; III, sperm incubated with A23187 (I, 57.43% $\pm$ 0.12%; II, 69.93% $\pm$ 0.84%; III, 73.7% $\pm$ 0.80%) (***$p < 0.001$).

capacitation and sperm motility compared with untreated sperm at the same duration of exposure.

We herein demonstrated that sperm underwent the acrosome reaction when exposed to EVs as evidenced by elevated ROS, indicating an increase in sperm capacitation. Mammalian sperm capacitation is an oxidative event, and the mitochondrial electron transfer chain precipitates the production of ROS in addition to its involvement in ATP synthesis, with ROS production occurs during tyrosine phosphorylation (*Rivlin et al., 2004*).

Although there appears to be an interaction between EVs and sperm, the mechanisms that mediate this communication remain unknown. The role of EV in intercellular communication has been studied in specific cells, and investigators have shown that

this communication can be mediated by cell membrane fusion or by direct binding to the cell surface (*Raposo & Stoorvogel, 2013*). Recent work has elucidated in detail the possible mechanisms underlying the internalization of EVs (*Aryani & Denecke, 2016*), such as cypermethrin-mediated endocytosis (*Barrès et al., 2010*; *Escrevente et al., 2011*), phagocytosis (*Feng et al., 2010*), macropinocytosis (*Escrevente et al., 2011*; *Tian et al., 2014*), and plasma (*Aryani & Denecke, 2016*; *Escrevente et al., 2011*; *Schwarz et al., 2013*) or endosomal membrane fusion. *Murdica et al. (2020)* used filipin-mediated inhibition to analyze the internalization of EVs by lipids, as filipin disrupts fossa structure and function as well as lipid raft-mediated endocytosis; internalization of EVs labeled with filipin was also shown to be strongly inhibited (*Rothberg et al., 1992*; *Rothberg et al., 1990*). These results showed that sperm internalized EVs through lipid raft structural domain-mediated endocytosis. Al-Dossary found that PMCA4a constitutes the $Ca^{2+}$ efflux pump in sperm, and that EVs are protein transporters from luminal fluid to the sperm surface. EVs thus transfer PMCA4a to sperm through the fusion of vesicles with the sperm membrane, regulating the dynamic balance of $Ca^{2+}$ (*Al-Dossary, Strehler & Martin-Deleon, 2013*).

We discerned that uterine fluid released EVs in the absence of external stimuli. These EVs were cup-shaped, 30–200 nm in diameter, and exhibited CD9 and CD63 proteins on their membranes; these features were also similar to those derived from EECs. Although the composition of EVs from uterine flushes is complex, the EVs more closely emulate the physiological state of the uterus. Additionally, although we used sperm from healthy male volunteers at the Center for Reproductive Genetics, we could not eliminate the possibility that normally active sperm might also manifest some infertility issues.

In summary, these results confirmed that EVs isolated from uterine fluid *in vitro* act on human sperm and promote their fertilizing ability. The increased capacitation after sperm interactions with EVs also suggests a possible physiological effect during the transit of the uterus. The exchange between sperm and uterine cells reveals a process by which sperm are prepared for fertilization of the oocyte, and this may have important implications for the treatment of human infertility.

## ACKNOWLEDGEMENTS

We would like to thank the staff of the Biobank of the First Affiliated Hospital of Kunming Medical University and Dr. Wenhui Lee's team from Kunming Institute of Zoology, Chinese Academy of Sciences, for their technical support.

### Funding

This research was funded by the National Natural Science Foundation of China (grant no. 82360294), the Yunnan Provincial Department of Science and Technology-Kunming Medical University Joint Special Fund for Applied Foundations Project (grant no.2014FB031), the Yunnan Provincial Department of Science and Technology-Kunming Medical University Joint Special Fund for Applied Foundations Key Project (grant no.

2019FE001[-005]), and the Yunnan Province Clinical Research Center for Chronic Kidney Disease (grant no. 202102AA100060). The APC was funded by the Yunnan Province Major Difficult disease (chronic renal failure) TCM and Western Clinical Cooperation pilot project. The funders had no role in study design, data collection and analysis, decision to publish, or preparation of the manuscript.

## Grant Disclosures

The following grant information was disclosed by the authors:
National Natural Science Foundation of China: 82360294.
Yunnan Provincial Department of Science and Technology-Kunming Medical University Joint Special Fund for Applied Foundations Project: 2014FB031.
Yunnan Provincial Department of Science and Technology-Kunming Medical University Joint Special Fund for Applied Foundations Key Project: 2019FE001[-005].
Yunnan Province Clinical Research Center for Chronic Kidney Disease: 202102AA100060.

## Competing Interests

The authors declare there are no competing interests.

## Author Contributions

- Renbin Deng conceived and designed the experiments, performed the experiments, analyzed the data, prepared figures and/or tables, and approved the final draft.
- Zhao Wu conceived and designed the experiments, prepared figures and/or tables, and approved the final draft.
- Chaoyong He performed the experiments, authored or reviewed drafts of the article, and approved the final draft.
- Chuncheng Lu performed the experiments, prepared figures and/or tables, and approved the final draft.
- Danpeng He analyzed the data, prepared figures and/or tables, and approved the final draft.
- Xi Li analyzed the data, prepared figures and/or tables, and approved the final draft.
- Zhenling Duan analyzed the data, authored or reviewed drafts of the article, and approved the final draft.
- Hui Zhao conceived and designed the experiments, authored or reviewed drafts of the article, and approved the final draft.

## Data Availability

 The raw measurements are available in the Supplementary Files.

## Supplemental Information

Supplemental information for this article can be found online at http://dx.doi.org/10.7717/peerj.16875#supplemental-information.

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
