# Peer review of "Exosomes from uterine fluid promote capacitation of human sperm"

_PeerJ, doi:10.7717/peerj.16875_

## Round 0.1 · original submission · Major Revisions

Dear authors.

Thank you for your submission. After reviewing the reviewers' comments, I request significant revisions before the manuscript can be considered further.

In addition, grammar and sentences are in great need of improvement to enable readers to better understand. Please proofread manuscripts before resubmission.

**Language Note:** The Academic Editor has identified that the English language must be improved. PeerJ can provide language editing services - please contact us at copyediting@peerj.com for pricing (be sure to provide your manuscript number and title). Alternatively, you should make your own arrangements to improve the language quality and provide details in your response letter. – PeerJ Staff

Reviewer 1 ·

Basic reporting

The manuscript was written in standard English language. However, the grammar and sentences need extensive improvements to provide a better understanding to the readers. The abstract need amendment as it does not include a concise description of the methodology and results. The figures need to be reviewed as the descriptions of statistical analyses were not included.

Experimental design

The experimental design was not clearly included (no sample size and criteria of the donor e.g. age etc). The methodology need to be written in passive voice.

Validity of the findings

No statistical analyses were included in the result section and in the Figures. The conclusion also needed some additional support statements as suggested in the attached pdf.

Additional comments

General comments:
Please ensure that the grammar is correct throughout the manuscript. Some clarifications need to be made to the materials and methods section and make sure that the graphs and figures are provided with concise descriptions of experimental groups and statistical analyses. Please check the in-text citation style for this journal.

Abstract
The abstract need amendment. Please include a concise description of the methodology and results.

Introduction
Line 26-28: Please rephrase the statement
- Any previous study on the contents of EVs in uterine fluid?

Methodology
Please describe the methodology in passive voice (e.g. in the western blot section). Other comments/suggestions are attached in the pdf.

Results
Figures: Please include a comprehensive statement to describe the data provided. Statistical analyses need to be included.
Other comments/suggestions are attached in the pdf.

Discussion
Line 179-180: Due to the rapid passage of sperm through the uterus, sperm are thought to be unaffected by the uterine environment. This statement contradicts with the following statements: Line 183-189: Early studies suggest that the uterus and oviduct may synergistically accelerate sperm capacitation (Hunter, 1969; Hunter and Hall, 1974). For example, it was found in wild boar that sperm could be temporarily stored in the uterus before entering the oviduct (Hunter and Rodriguez-Martinez, 2004). fusi, FM. et al demonstrated that incubation of endometrial epithelialcells with sperm would enhance capacity. The number of sperm penetrating hamster oocytes was also significantly increased after co-culture of endometrial epithelial cells and sperm (Fusi,Viganò, Daverio et al., 1994). Consider revising the statement.
Line 199-201: In our experiments, there was no significant change in sperm motility after capacitation, considered due to insufficient incubation time. We believe that there are significant individual differences in sperm. capacitation, related to the quality of sperm. (therefore I suggest to include a reference why 2 hr incubation time was used in this study)

Annotated reviews are not available for download in order to protect the identity of reviewers who chose to remain anonymous.

Reviewer 2 ·

Basic reporting

The researchers in this study would like to highlight the importance of EVs extracted from the uterine fluid and their physiological role in the sperm capacitation process.
Some consideration:
1. The fertilization process occurs in a fallopian tube, is there any significance or presence of EVs in the fallopian tube fluid too? Have the authors tried to check how capacitation can be facilitated and/or completed in fallopian tubes?
2. Capacitation process starts from ejaculation into the cervical fluid, would there be any implications if the sperms were not subjected to cervical fluid but rather directly to uterine EVs? As mentioned in the methodology, to prevent capacitation during the incubation, human serum albumin and NaHCO3 were omitted from the medium, but how do we assure this? Any evidence to support this?

Experimental design

The experimental design is not properly highlighted in the manuscript.

1. Use of past tense in methodology- grammar check for example first line of Western blot: Exosomes were extracted and subjected to protein quantification by BCA.
2. Please mention the justification for using only two exosome markers for WB. Better to mention the names of antibodies, manufacturers, and target molecular weight/Kd in the methodology. The significance can be highlighted in the result section below to figure legend with statistical values.
3. Use of past tense in methodology- grammar check for example first line of Western blot: Exosomes were extracted and subjected to protein quantification by BCA.
4. Can authors draw a flow diagram to show 3 groups of sperms that were incubated with EVs, (compared with two controls)?

Validity of the findings

1. Please mention the justification for using only two exosome markers for WB. Better to mention the names of antibodies, manufacturers, and target molecular weight/Kd in the methodology. The significance can be highlighted in the result section below to figure legend with statistical values.
2. Western blot images (uncropped) also do not show protein ladder and standard. Why?
3. What is the housekeeping gene used in WB? If it has not been used, mention the reason.
4. What are labels 1, 2, and 3 in WB?? The row shows more samples loaded in comb wells.

5. Statistical analysis stated that experiments were repeated 3 times. Do the experiments repeat with different subject samples or the same? How many samples were collected in total and why only 3 times it is repeated? What was the standard deviation? Not mentioned.
6. Transmission Electron microscopy images A and B are showing EVs. What is the difference between A & B? Scale bars are not clearly visible, at what magnification it is taken. Also, C and D in same figure show plots for particle and diameter. Diameter/nm is understandable but what is the significance of particles/ml? Can you elaborate?
7. The concentration and average size of nanoparticles in each sample were calculated and presented with standard error. (What was the standard error? Not mentioned in the figure legend.
8. For acrosomal reaction, 3 groups are used. Why were the same groupings not used for ROS measurements?
9.. Why is the measurement of ROS only done in sperms subjected to EVs? The comparison is done only with one negative control group. Is there a possibility to add positive control? Could add a few points for DCFH-DA use in flow cytometry. For example, to detect ROS.
10. Curve velocity and Sperm motility graph is not clear and very blurry.
11. Sperm heads with intact acrosomes were blue, and the regions of the sperm heads after the reaction were unstained (H. Y. Lu, Lu, Hu et al., 2002). (Figure 2) Figure 2 provided a western blot. most probably it is Figure 4. that needs to be corrected.
12. For Figure 5A in the result section, please clearly state what control was used. One was negative control (without EVs, and the other was positive control.
13. Which result data shows elevated ROS fluorescent intensity? The figure number is not stated in the manuscript. It should be properly highlighted in the result section with the corresponding figure number. What is the significance of A, B, C, and D in flow cytometry? Elaborate.

Additional comments

The sections on methodology and results need to be revised thoroughly.
Better to draw a diagram or illustration for the methodology and techniques used.
In the section on results: Figure legends MUST be provided with details in the text of the manuscript. What are control and compared groups? In each figure clearly state, what is A, B, C, and D. Also, statistical significance values to whom (comparison) along with standard deviations must be quoted. Figure numbers in the manuscript text are different as compared to figure labels in files.
Below the figure legend, provide a brief text explanation for the results obtained. Figures are not clear and are blurred. Provide uncropped western blot with ladder indicating mw.
Grammar and sentence construction should be re-checked.

---

## Round 0.2 · Minor Revisions

Dear authors,

Thank you for resubmitting your manuscript.

Please revise your manuscript according to the reviewer's comments.

Reviewer 2 ·

Basic reporting

Abstract:
Row 20. Please add aim of this study (one sentence).
Row 27. The first sentence is not needed. Already mentioned in row 22 methods. In the same row 22 “The fraction isolated from the uterine fluid was observed cup-shaped vesicles of different sizes by TEM” can add EVS before fraction.
Row 30 and 31: Grammer and sentence corrections to be done such as “Comparing the motility of the two groups incubated sperm motility significantly differed at 4h”. 31 incubated also please check correct verb tense.
In abstract conclusion: Also sentence construction and grammar need to be corrected.
Methodology:
Row 108 the font color of word is different.
Discussion:
Row 246, please provide full form of EEC.
Row 234-235 reference provides different spelling of same author. Please keep reference either at the start of sentence or at the end.
Row 238-239: sentence construction!
Row 240 reference correction same as row 235.
From row 240 to 243, could you please elaborate in simple, small sentences.
Row 250: It is unclear what authors like to highlight, the sperm as well as uterine flushing (EVs) were obtained from healthy adults. This paragraph needs a clear statement.
Row 420: Figure 3. The figure legend is repeating the methodology stated earlier. Please state A represents sperms motility after 2hrs incubation with EVS. B represents sperms motility after 4hrs incubation with EVS. Blank: Sperm didn’t incubate with EVs. EVs group: Sperm incubated with EVs. Three biological replicates were performed. The data represented the mean and SEM of the three experimental data. A: 51.63%±1.32% vs 52.10%±1.51%, P>0.05. B:49.3%±0.84% VS 55.37%±0.73%, **P<0.01.
The results showed that sperm motility was no significant difference at 2h and showed that sperm motility was significantly difference at 4h. (sentence construction need to be looked here).
In Figure 4 legend, it is better to bring A, B and C at row 403.
Row 439: Figure 5. You may just start with A, B and C description. The sperms were divided in to three groups is not needed to be mentioned in figure legend.
Row 449. Figure 6. Same. You may just start with A, B and C description. The sperms were divided in to three groups is not needed to be mentioned in figure legend.

Experimental design

The authors have done considerable revision and to elaborate experimental design. They have added controls and comparison groups. This is acceptable and clear.
Suggestion: It would have been even better if the sperm from healthy donors and asthenozoospermia patients were incubated with EVs to assess EVs potential.
As authors stated in the discussion, there were many studies conducted previously on the sperm internalization by EVs, this study aim was to elucidates sperm capacitation mainly.

Validity of the findings

The findings seem to be acceptable. However, it was mentioned that experiments were repeated three times with each (three) sperm donor samples. Was it repeated with different EVs obtained from 10 uterine douche samples? The aim of this study was to analyze sperm paraments and not EVs mechanisms which has already been studies in the literature. Addition of many sperm donor samples could have produced more authenticity.

Additional comments

In many parts of the manuscript, sentence construction and grammar should be re-checked. Strongly advise to proofread the manuscript.

---

## Round 0.3 · accepted · Accept

The authors have revised the manuscript accordingly.

Reviewer 2 ·

Basic reporting

All comments have been addressed by the authors.

Experimental design

Good

Validity of the findings

All analysis is in the supplemental files provided. Uncropped clear western blot can also be seen. OK

Additional comments

The authors have revised the manuscript and have addressed all concerns and comments. The paper is acceptable at this stage.